# Effects of the Blending Ratio on the Design of Keratin/Poly(butylene succinate) Nanofibers for Drug Delivery Applications

**DOI:** 10.3390/biom11081194

**Published:** 2021-08-12

**Authors:** Giulia Guidotti, Michelina Soccio, Edoardo Bondi, Tamara Posati, Giovanna Sotgiu, Roberto Zamboni, Armida Torreggiani, Franco Corticelli, Nadia Lotti, Annalisa Aluigi

**Affiliations:** 1Department of Civil, Chemical, Environmental, and Materials Engineering, University of Bologna, Via Terracini 28, 40131 Bologna, Italy; giulia.guidotti9@unibo.it (G.G.); m.soccio@unibo.it (M.S.); edoardo.bondi2@studio.unibo.it (E.B.); 2Institute of Organic Synthesis and Photoreactivity, Italian National Research Council, Via P. Gobetti 101, 40129 Bologna, Italy; tamara.posati@isof.cnr.it (T.P.); giovanna.sotgiu@isof.cnr.it (G.S.); roberto.zamboni@isof.cnr.it (R.Z.); armida.torreggiani@isof.cnr.it (A.T.); 3Kerline srl, Via Piero Gobetti 101, 40129 Bologna, Italy; 4Institute for Microelectronics and Microsystems, National Research Council, Via P. Gobetti 101, 40129 Bologna, Italy; corticelli@bo.imm.cnr.it

**Keywords:** keratin, poly(butylene succinate), electrospinning, drug release

## Abstract

In recent years there has been a growing interest in the use of proteins as biocompatible and environmentally friendly biomolecules for the design of wound healing and drug delivery systems. Keratin is a fascinating protein, obtainable from several keratinous biomasses such as wool, hair or nails, with intrinsic bioactive properties including stimulatory effects on wound repair and excellent carrier capability. In this work keratin/poly(butylene succinate) blend solutions with functional properties tunable by manipulating the polymer blending ratios were prepared by using 1,1,1,3,3,3-hexafluoroisopropanol as common solvent. Afterwards, these solutions doped with rhodamine B (RhB), were electrospun into blend mats and the drug release mechanism and kinetics as a function of blend composition was studied, in order to understand the potential of such membranes as drug delivery systems. The electrophoresis analysis carried out on keratin revealed that the solvent used does not degrade the protein. Moreover, all the blend solutions showed a non-Newtonian behavior, among which the Keratin/PBS 70/30 and 30/70 ones showed an amplified orientation ability of the polymer chains when subjected to a shear stress. Therefore, the resulting nanofibers showed thinner mean diameters and narrower diameter distributions compared to the Keratin/PBS 50/50 blend solution. The thermal stability and the mechanical properties of the blend electrospun mats improved by increasing the PBS content. Finally, the RhB release rate increased by increasing the keratin content of the mats and the drug diffused as drug-protein complex.

## 1. Introduction

Nowadays, one of the main challenges facing biomedicine is the need to be able to administer drugs and active ingredients in a controlled and targeted manner. Therefore, precision design principles have been recently adopted to develop new methods of administration, which can be varied according to the treatment required, and limit any side effects as much as possible.

Currently, in the wide spectrum of biomolecules used for biomedical applications, natural proteins such as gelatin, collagen, keratin or silk fibroin are fascinating biomolecules owing to their availability and favorable properties such as biocompatibility, biodegradability and capability to mimic the in vivo environment of the human body [1]. Among the natural proteins, keratin is the most abundant one, being the major component of hair, wool, nails, hooves, claws, scales, horn, beaks and feathers, and it is also found in epithelial cells [2]. The regenerative and neuroinductive properties of keratin have been attributed to the presence of numerous Leu-Asp-Val (LDV), cell adhesion sequences as well as glutamic acid-aspartic acid-serine (EDS) [3,4]. Moreover, the primary structure composed of both hydrophilic and hydrophobic aminoacids makes this protein a good carrier of different kinds of active ingredients (with both hydrophilic and hydrophobic characteristics, as well as negatively or positively charged), thereby representing an interesting biomolecule for drug delivery purposes [5].

However, the processing of keratin into structured materials is very challenging. Indeed, this protein is characterized by relatively low molecular weights and wide molecular weight distributions which confer keratin solutions with poor rhological properties for their processing. The blending of keratin with other both natural or synthetic biopolymers is a common strategy used to overcome the aforementioned drawback and to improve its processing in order to fabricate biomaterial systems and devices with tunable chemical, mechanical, biological and biomimetic properties according to the desired application [6,7]. As to synthetic polymers, their easy synthesis, processability and the ability to tune their properties in view of different uses, make them particularly suitable for a wide plethora of applications.

As examples, processing keratin solutions blended with other polymers e.g., poly-ethylene-oxide (PEO) [8], polyvinyl alcohol (PVA) [9] or polycaprolactone (PCL) [10] by electrospinning, has been widely explored as s valuable platform for controlled drug delivery. The main advantages attributed to electrospun mats over conventional topical drug delivery systems (like transdermal patches containing gels or liquid sprays) are a large surface area-to volume ratio, enhanced flexibility, high drug encapsulation efficiency and nanostructured morphology mimicking the extracellular matrix [11]. Electrospun membranes also provide high occlusive effects, while bringing at the same time great breathability to the skin because of their high porosity [12]. Moreover, the encapsulation of drugs within the nanofibers provides more stability to the drug and prevents its crystallization [13].

Within this framework, recently proposed [6] electrospun mats made of keratin and poly(butylene succinate) (PBS) represent a valuable platform for drug delivery that deserves to be studied in more depth. PBS is a bio-based aliphatic polyester, which has recently obtained U.S. Food and Drug Administration (FDA) approval and has received a great deal of attention for biomedical applications due to its biocompatibility, biodegradability and good capability of binding to natural fibers [14,15,16,17]. In addition, its good heat resistance and melting temperature, which is one of the highest among aliphatic polyesters, provide a wide processing range. In a previous work, PBS was blended with keratin at a 50:50 blending ratio using 1,1,1,3,3,3-hexafluoroisopropanol (HFIP) as common solvent. Despite their immiscibility, the blend solutions were successfully electrospun into nanofibrous mats with good mechanical properties due to the presence of PBS and with stimulatory effect towards fibroblast growth induced by the presence of keratin [6].

This work is the continuation of the previous one, since the keratin/PBS blend solutions with different polymer blending ratios are electrospun in order to obtain nanofibers with functional properties tunable by acting on the polymer mixture composition. To this end, the stability of keratin in HFIP was studied by comparing the molecular weight distribution of the pristine protein with that of the protein regenerated from the solvent. Moreover, the relationships between keratin/PBS blending ratios and the rheological behavior of the processing solutions as well as the mechanical, thermal and adhesion properties of the electrospun mats were investigated. Finally, in order to understand the potential of such mats as drug delivery systems, electrospun keratin/PBS blend nanofibers loaded with rhodamine B (RhB) were prepared and the drug release mechanism and kinetics as function of blend composition was studied. RhB was chosen as model drug since it is easily detectable by UV-visible measurements.

## 2. Materials and Methods

### 2.1. Materials

High molecular weight keratin powder (~50 kDa) extracted from raw wool was kindly donated by Kerline Srl (Bologna, Italy). Poly(butylene succinate) (PBS), with a molecular weight (M_n_) of 50,000 g/mol, was synthesized from dimethyl succinate (DMS) and 1,4-butanediol (BD) as described in a previous work [6]. The materials for electrophoresis were purchased from BioRad Laboratories, Inc (Hercules, CA, USA), while all other chemicals were purchased by VWR Chemicals (VWR International LLC, Radnor, PA, USA).

### 2.2. Keratin Films Regenerated from 1,1,1,3,3,3-Hexafluoro-Isopropanol (HFIP)

In order to prepare keratin films regenerated from HFIP, pristine keratin was first dissolved overnight in HFIP at a concentration of 13% *w/v* that has been previously established as the best concentration for the electrospinning of keratin (see Appendix A). After that, the solution was cast in a glass mould and the solvent was allowed to evaporate in oven for 24 h at 40 °C.

### 2.3. Electrospinning of Keratin-PBS Solutions Loaded with RhB

The solutions for electrospinning were prepared first by separately dissolving PBS (520 mg) and keratin (520 mg) in HFIP (4 mL) containing RhB (3.9 mg/mL), in order to obtain solution at a polymer concentration of 13% *w*/*v*. The two solutions were maintained overnight under magnetic stirring, covering the keratin solution with an aluminum foil to avoid the exposure of RhB to light. After that, the two solutions were mixed together, at different blending ratios (KER/PBS: 30/70, 50/50 and 70/30), and left under stirring for at least one additional hour, still covered. The final solutions were then loaded into a 5 mL syringe connected to a needle (0.8 mm internal diameter) and electrospun towards a static collector (10 × 10 square centimeters) using an applied field of 1 kV/cm (applied voltage 18 kV, needle-collector distance 18 cm) and a flow rate of 1.8 mL/h. For each kind of sample, three production processes were repeated in order to validate the reproducibility of the process.

### 2.4. Characterization

The molecular weight distribution of keratin powder and keratin regenerated from HFIP, were analyzed by sodium dodecyl sulphate-polyacrylamide gel electrophoresis (SDS-PAGE), using a Mini-PROTEAN^®^ Tetra Vertical Electrophoresis Cell (BioRad Laboratories, Inc., Hercules, CA, USA). The samples were dissolved in a buffer solution of Tris-Hcl (500 mM, pH 8.6) containing DTT (140 mM) and urea (8 M). These keratin solutions were added to the 2XLaemmli sample buffer in a 1:1 ratio and boiled at 90 °C for 5 min in order to denature the protein. The separation was performed in a 4–20% gradient gel at 200 V for 45 min by using 20 μL of sample solution (10 μg/μL), for each well. When the separation was finished, the gel was rinsed with ultrapure water and stained overnight with a Coomassie Brilliant Blue R-250 staining solution. Destaining was done overnight in diluted acetic acid (5%).

Rheological properties of the polymer blends solutions were determined carrying out shear rate-dependent viscosity measurements, using a MCR 102 Compact Rheometer (Anton Paar GmbH, Graz, Austria) equipped with a PDT 200/56/l Peltier temperature control device, set at 25 (±0.1) °C, using a cone plate geometry (75 mm diameter, 1° angle and 45 μm truncation), in controlled shear rate mode. The shear rate was ranged from 0.1 to 1000 s^−1^. Data were acquired and elaborated with the RheoCompass Software (Anton Paar).

The electrospun nanofiber morphology was investigated by scanning electron microscopy (SEM) using an EVO LS 10 LaB6 scanning electron microscope (Carl Zeiss Microscopy GmbH, Oberkochen, Germany) with an acceleration voltage of 5 kV and working distances of 4.3 mm and 5.2 mm, respectively. Samples were gold-sputtered for 1 min before the analysis. The fiber diameters were obtained by using GIMP software (National Institutes of Health, Bethesda, MD, USA). Two SEM images of the samples obtained from each production process were acquired. The mean diameter and the diameter distribution of nanofibers were evaluated from 150 measurements randomly gathered from all the acquired SEM images for each sample.

Infrared spectra were acquired using the attenuated total reflectance technique (ATR) with a Vertex 70 interferometer (Bruker Corporation, Billerica, Massachusetts, USA) equipped with a diamond crystal single reflection platinum ATR accessory, in the 4000–600 cm^−1^ region, with 26 scans and a resolution of 4 cm^−1^. A TGA4000 instrument (Perkin Elmer Corporation, Waltham, MA, USA.) was used to carry out thermogravimetric analysis (TGA). Measurements were performed heating each sample (≈5 mg) at 10 °C/min from 40 to 800 °C, under pure nitrogen flux (40 mL/min). The temperature of initial degradation (T_id_) and the one corresponding to maximum degradation rate (T_max_) were calculated.

In order to determine the main thermal transitions, i.e., glass transition temperature (T_g_), denaturation/degradation temperature (T_d_) and melting temperature (T_m_), together with the heat associated to these transitions, calorimetric analysis was performed by means of a Perkin Elmer DSC6 instrument, calibrated with indium and zinc standards. Weighed samples of about 10 mg were subjected to a I scan by heating at 20 °C/min from −70 °C to about 40 °C above melting temperature in case of PBS, up to 380 °C for the samples containing keratin.

Tensile tests were performed using a 5966 machine (Instron^®^, Norwood, Massachusetts, USA) equipped with a 1 KN load cell. At least six rectangular specimens (5 × 50 suqre millimeters) were tested for each electrospun mat, adopting a 10 mm/min crosshead speed and a gauge length of 20 mm. Stress-strain curves were obtained from load- displacement data directly measured by the software. Stress (σ_B_) and strain (ε_B_) at break were calculated, together with elastic modulus (E), this last extrapolated from the initial linear slope of the stress-strain curves.

Adhesion tests were carried out at room temperature on electrospun mats obtained from PBS and from Keratin/PBS 50/50 blend, using a cellulose acetate membrane to simulate the human skin. More in details, rectangular strips with dimensions of 5 × 50 square millimeters were obtained from both electrospun mats and cellulose acetate membranes. Electrospun strips as well as cellulose acetate ones were immersed in distilled water for one minute, and dripped on filter paper, to simulate the humidity of human skin. Then, they were overlapped each other for a length of 2 cm. The so-obtained specimens, 6 for each sample, were then stretched at a constant speed (10 mm/min). The mean value of shear stress (σ_S_), corresponding to maximum stress reached before the mat and cellulose acetate detached from each other, was measured. The same tests were also conducted in dry conditions: the specimens were prepared using the procedure described above, but they were stored at room temperature until complete drying before proceeding with the measurements.

### 2.5. Drug Release Test

To determine the drug release profiles from nanofibrous membranes, 1 square piece (2 × 2 square cenimeters) was cut from each sample loaded with RhB and put in 6 mL of phosphate buffer with a pH of 7.4 and a temperature of 37 °C. Aliquots of 100 μL were removed at the specific time intervals and replaced with fresh buffer; the released RhB was detected at 554 nm using a UV-Vis spectrophotometer (Cary 100—Agilent Technologies, Santa Clara, CA, USA), after the determination of a calibration curve (as described in the Appendix A). For each sample, at least five drug release tests were carried out in order to determine the mean drug release profile. The Korsmeyer-Peppas mathematical model was used to determine the release mechanism and kinetic of RhB from the electrospun membranes.

## 3. Results and Discussion

### 3.1. Keratin Regenerated from HFIP

The stability of keratin in HFIP was investigated by analyzing the protein molecular weight distribution by SDS-PAGE analysis. In Figure 1 the electrophoretic pattern of pristine keratin (line 1) was compared with that of keratin regenerated from HFIP (line 2). The pristine protein shows the characteristic bands at 60 and 45 kDa attributed to the low sulphur keratin and the bands at 28 and 11 kDa related to the high-sulphur keratin [18]. The bands that can be observed at molecular weights higher than 60 kDa are attributed to the S-S cross-linked keratin chains. Keratin regenerated from HFIP shows the identical pattern of pristine one confirming that the solvent does not degrade the protein.

### 3.2. Rheological Behavior of Keratin/PBS Blend Solutions

The rheological behavior of the immiscible system made of keratin and PBS blend solutions in HFIP was investigated through steady shear measurements.

In Figure 2, the apparent viscosity (*η*) is plotted as function of shear rate ($\dot{\gamma }$). A shear thinning behavior was observed for all the tested blending ratios. The shear-thinning is caused by the disentanglement of the polymer coils in solution and/or increased orientation of the polymer coils in the direction of the flow. For each blending ratio, the shear thinning region was fitted with the Carreau equation (Equation (1)):(1)$$\eta =\eta_{\infty }+\frac{\eta_{0}-\eta_{\infty }}{\;[1+{{\left(\lambda_{c}\dot{\gamma }\right)}^{2}]}^{n}}$$
where *η*_0_ (Pa s) is the zero-shear ${\dot{\gamma }}_{t}$ viscosity, the $\eta_{\infty }$ (Pa s) is the infinity-shear viscosity, the $\lambda_{c}$ (s) is a time constant related to the relaxation times of the polymer chains in solution and *n* is the power law flow behavior index. The rheological parameters and the correlation coefficients (R^2^) of the fitting are shown in Table 1.

As shown, for all the blending ratios, the *n* values are lower than 1 indicating that all the blend solutions exhibited well-behaved shear thinning properties [19].

Among the Keratin/PBS blend solutions, the 50/50 blend showed the highest zero-shear viscosity and the lowest infinite-shear viscosity, thereby representing a ${\dot{\gamma }}_{t}$ more pronounced shear-thinning behvior. In addition the 50/50 blend also showed the lowest value of the time constant $\lambda_{c}$. This time constant $\lambda_{c}$ provides an effective indiator of the onset shear rate for the shear thinning ${\dot{\gamma }}_{t}$, that is in turn the critical value of shear rate for which the applied flow overcomes the Brownian forces tending to bring the polymer chains into the isotropic conformation, and the polymer chains orient themselves in the direction of the applied stress [20]. The lowest $\lambda_{c}$ corresponds to the the highest ${\dot{\gamma }}_{t}$, suggesting a greater attitude of the polymer chains in the Keratin/PBS 50/50 blend solution to relax into the isotropic conformation.

### 3.3. Morphology of Keratin/PBS Blend Nanofibers Loaded with RhB

All the keratin/PBS blend solutions doped with RhB were successfully electrospun into free standing membranes as shown in Appendix A. Figure 3a shows the SEM images of the RhB loaded keratin/PBS blend nanofibers and the related diameter distributions. All the randomly oriented nanofibers are cylindrical in shape and defect-free, since no beaded or ribbon-like fibers were observed (Appendix A). The diameters distributions showed in Figure 3b, as well as the mean diameters and the relative diameters distribution were discussed to evaluate dimension and homogeneity of the blend nanofibers. As shown compared to the Keratin/PBS 30/70 and 70/30, the blend 50/50 gives rise to bigger and less homogenous nanofibers, showing a broader diameter distribution. Probably, the greater resistance of the polymer chains to be aligned along the applied flow (lowest $\lambda_{c}$) contributes to contracting the polymer jet, thus reducing the chains’ elongation during electrospinning and resulting in less homogenous and thicker nanofibers. Instead, the Keratin/PBS 70/30 and 30/70 blend solutions give rise to nanofibers with a narrower diameter distribution. In this case, the enhanced tendence of the polymer chains to be aligned along the direction of the applied shear (higher $\lambda_{c}$) contributes to promote and maintain the elongation of polymer chains during the shear thinning imposed by electrospinning process.

### 3.4. Mechanical Properties and Skin Adhesion

Tensile tests have been carried out only on PBS, Keratin/PBS 30/70 and Keratin/PBS 50/50 electrospun mats. Indeed, although the blend richest in keratin (Keratin/PBS 70/30) was electrospinninable, the obtained electrospun mat was too fragile to be subjected to these tests with the tools provided. The values of elastic modulus (E), stress (σ_B_) and elongation (ε_B_) at break are reported in Table 2, and the relative stress-strain curves are shown in Figure 4a. As expected from previous studies [6], the samples are characterized by comparable values of elastic modulus and stress at break, being in both cases slightly lower for the keratin/PBS blend electrospun mats with respect to PBS. This result confirms that it was possible to obtain processable mats, whose stiffness was comparable to that of PBS and significantly lower than the one of keratin. Instead, elongation at break of PBS electrospun mat was almost twice the value of Keratin/PBS 30/70 and triple of Keratin/PBS 50/50 one. These lower elongation at break values can be ascribed to the immiscibility between keratin and PBS. Obviously, in the case of the co-continuous phase of the Keratin/PBS 50/50 blend, the more extended interphase zones where repulsive interactions occur between the two polymers contribute to further reduce the elongation at break.

In order to check the adhesion ability of the materials under investigation, in vitro adhesion tests on PBS and Keratin/PBS 50/50 mats were performed, in both wet and dry conditions, using membranes of cellulose acetate as synthetic skin model [20]. The values of shear stress (σ_S_) are reported in Table 2 and represented in Figure 4b. PBS mat adhered more difficultly to cellulose acetate than the one containing keratin. This behavior was confirmed by the σ_S_ values (0.7 MPa for PBS vs. 1.5 MPa for Keratin/PBS 50/50): the shear resistance was therefore more than doubled in the case of the blend, thanks to the very high adhesion conferred by keratin. Moreover, the detachment of keratin/PBS 50/50 electrospun mat occurs after a slight elongation, indicating that this material is more prone to deform rather than detach from the synthetic skin (Appendix A). Adhesion on completely dry specimens was also verified. In these conditions the two samples show a completely different behavior: while PBS, after drying, detached from the cellulose acetate membrane, Keratin/PBS 50/50 blend still adhered. The shear stress value for this dry mat (σ_S_ *) although slighlty lower than the one obtained in wet conditions, is a further proof of how keratin contributes to enhance the adhesion properties of the blend electrospun mat.

### 3.5. Thermal Behavior

The thermal stability of keratin, PBS and their blend nanofibers has been analyzed by TGA measurements. The temperatures of initial degradation (T_id_) and of maximum weight loss rate (T_max_) are collected in Table 3, while the corresponding thermograms are reported in Figure 5a. As known, PBS turned out to be the most thermally stable, while keratin the lowest [6]. As to the keratin degradation profile, weight loss occurs at 71 and 188 °C, these jumps ascribable to evaporation of water bounded with keratin through physical interactions and hydrogen bonds, respectively, as well as at 271 °C, due to the protein denaturation/degradation [21].

Conversely, PBS shows a 2-steps weight loss, the main one starting at 363 °C. Keratin/PBS blends show an intermediate behaviour, their thermograms showing multiple degradation steps, as a proof of the improving effect of PBS in terms of thermal stability.

DSC curves (I scan) are shown in Figure 5b, while the corresponding thermal transitions are listed in Table 3. PBS shows the typical profile of semicrystalline polymers, with a slight baseline deviation at −32 °C, ascribable to glass to rubber transition, followed by an endothermic phenomenon at 114 °C due to the melting of crystalline portion of the polymer.

Conversely, keratin shows three endothermic phenomena, which differ both in shape and intensity: the first one, located at 88 °C, is related to evaporation of water bounded with the protein, while the second at 226 °C, is attributed to the denaturation secondary structures of keratin, in particular α-helix and the one at higher temperature (282 °C), ascribed to the protein degradation in agreement with TGA data. The DSC traces of the mats containing both PBS and keratin show both the phenomena characteristic of pure keratin (T_d_) and the transitions typical of PBS (T_g_ and T_m_), with intensities consistent with the composition of the blends.

### 3.6. Drug Release

Herein, the RhB release from Keratin/PBS 70/30 and 50/50 blend electrospun mats was evaluated as a function of the blend composition and fiber morphology. Keratin/PBS 30/70 electrospun mat was not considered due to the difficulties encountered in obtaining reproducible drug release profiles. The profiles of RhB release from Keratin/PBS 70/30 and 50/50 electrospun mats are presented in Figure 6. As seen, the 90% of RhB was released from Keratin/PBS 70/30 sample within the first hour (Figure 6a), while in the following 50 h (2 days) a slow release up to 100% took place (Figure 6b). Also for Keratin/PBS 50/50, a biphasic RhB release can be seen, but in this case the burst release was significantly reduced to 56% in the first hour and to 75% in the following 2 days. The observed behavior related to the drug release is very attractive for the engineering of a patch obtained from a suitable integration of the both electrospun mats with a drug release profile desirable for wound healing that has to be faster within the first hours of treatment (60%) and sustained for long periods (releasing >90% of the loaded drug, from 18 h to 5 days). The benefits for this kind of release rely on the need of smaller amount of drug within time, once the wound healing process advances.

To further study the mechanism of the biphasic release of Rhb from Keratin/PBS blend electrospun mats, the RhB release profiles in the first hour of release were analyzed by the Kersemeyer-Peppas model (Equation (2)):(2)$$Q_{t}=kt^{n}$$
where $Q_{t}$ is the drug fraction released at time *t*, *k* is a constant reflecting the structural and geometric characteristics of the fibers and *n* is the release exponent, which indicates the drug release mechanism. The obtained results are shown in Figure 6a and the obtained parameters are shonw in Table 4. According to the regression coefficient R^2^ values, a good fit was observed for both the blend nanofibers. In addition, the obtained *n* values were less than 0.45, indicating that Rhb release from both samples followed the Fickian diffusion mechanism. Moreover, the kinetic constant of RhB release from Keratin/PBS 70/30 sample is higher than that obtained from Keratin/PBS 50/50, suggesting a faster drug release from the keratin-richer nanofibers [22].

In order to better understand the polymeric matrix behaviour during RhB release, FTIR spectra of the samples before and after the release tests were compared (Figure 7).

In Figure 7, the FTIR spectra of Keratin/PBS 50/50 and 70/30, pre- and post-release and normalized for the C=O adsorption peaks of PBS are shown (1715 cm^−1^). The FTIR spectra of the starting samples (pre-release) show the typical adsorption peaks associated to the amide I (1650 cm^−1^) and amide II (1590 cm^−1^) of keratin [6]. As it can be observed, in the spectra acquired after drug release test, the keratin peaks have totally disappeared, suggesting that the drug is probably released as a drug-protein complex through a Fickian diffusion mechanism.

## 4. Conclusions

In this study, keratin/PBS blend solutions with different blending ratios were successfully electrospun into nanofibrous mats using HFIP as common solvent. All the blend solutions displayed a shear thinning behavior which is more pronounced in the Keratin/PBS 50/50 blend solution. Moreover, the Keratin/PBS 30/70 and 70/30 blend solutions displayed a higher attitude of the polymer chains to orient along the applied shear, giving rise to nanofibers with narrower diameter distributions. The addition of PBS contributed to improve the mechanical properties of the blend electrospun mats, making possible to obtain free-standing and processable mats, even if it has been impossible to measure the mechanical parameters of the sample richest in keratin. Conversely, compared to PBS electrospun mat, the Keratin/PBS 50/50 blend one was characterized by a better adhesion to cellulose acetate membranes, used to mimic human skin, both in wet and in dry conditions, confirming the unaltered adhesion capability of keratin in the blend. The thermal stability of the blend nanofibers ranged from those of keratin and PBS, and it increased by increasing the PBS content. Furthermore, the RhB release strongly depended on the nanofibers’ composition. In particular, a faster RhB release was obtained for the samples richer in keratin. Finally, the release occurred through a drug-protein complex and by means of a Fickian diffusion mechanism.

The results obtained for the single electrospun mats are meaningful for the design of patches with desired and tailored properties such as thermal and mechanical behavior, adhesion properties as well as drugs loading and drug release profiles. These patches can be engineered simply by exploiting the additive manufacturing of the electrospinning process through a layer-by-layer deposition of electrospun mats.

## Figures and Tables

**Figure 1 biomolecules-11-01194-f001:**
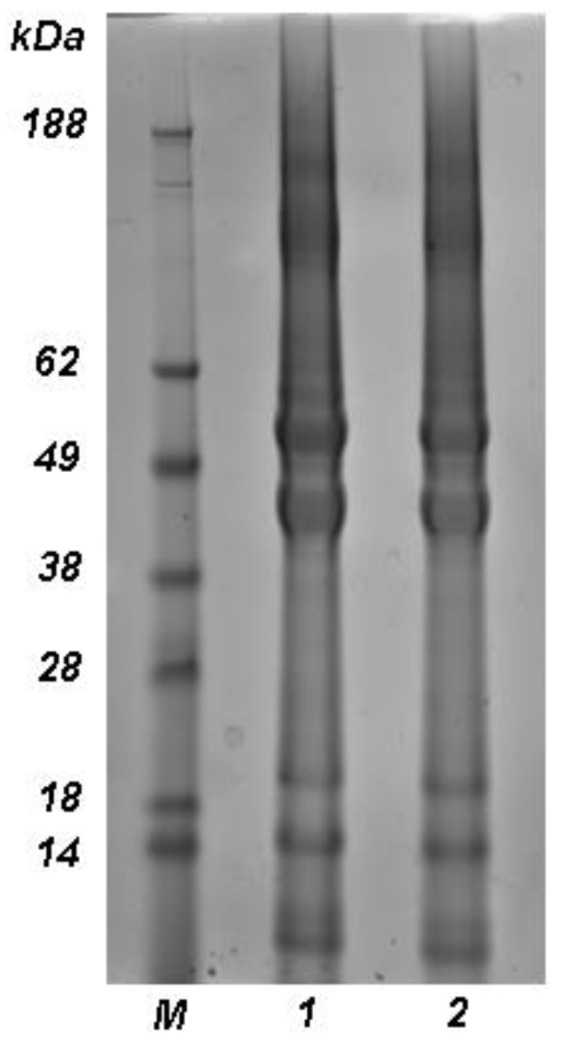
Electrophoresis pattern of marker (**line M**), pristine keratin (**line 1**) and keratin regenerated from HFIP (**line 2**). The electrophoresis was carried out using a 4–20% gradient gel, by loading each well with 20 μL of sample solution (10 μg/μL).

**Figure 2 biomolecules-11-01194-f002:**
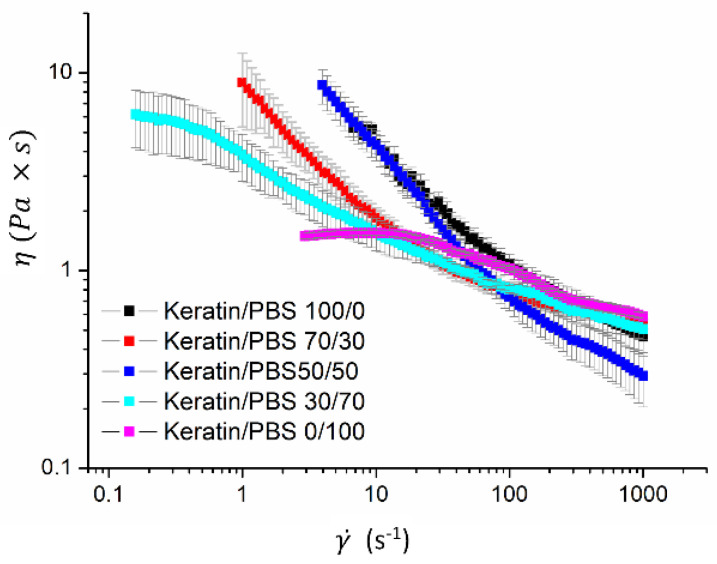
Apparent viscosity (*η*) as function of shear rate ($\dot{\gamma }$) of keratin/PBS blend solutions at different blending ratio.

**Figure 3 biomolecules-11-01194-f003:**
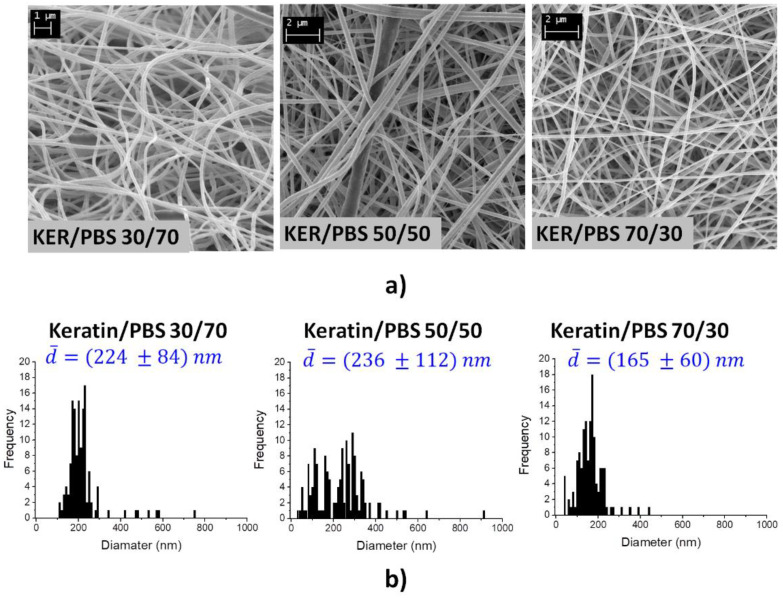
(**a**) SEM images and (**b**) Diameter distribution and mean diameter ($\bar{d}$) of keratin/PBS 30/70, 50/50 and 70/30 blend nanofibers.

**Figure 4 biomolecules-11-01194-f004:**
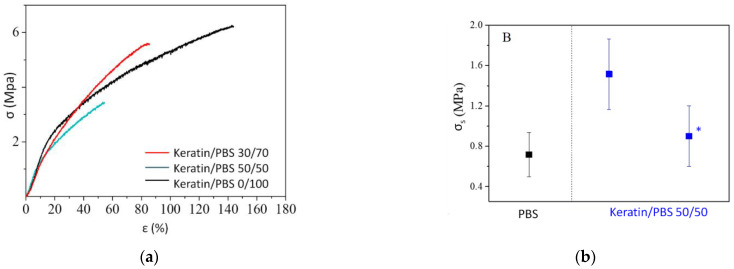
(**a**) Stress-strain curves and (**b**) shear stress values (σ_S_) of PBS and keratin/PBS electrospun mats in wet and (*) dry conditions.

**Figure 5 biomolecules-11-01194-f005:**
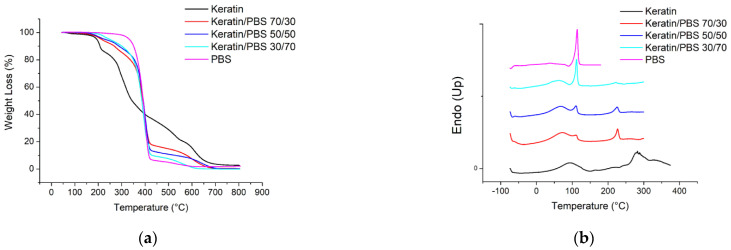
(**a**) TGA and (**b**) I scan DSC curves of PBS, Keratin and Keratin/PBS blend electrospun mats.

**Figure 6 biomolecules-11-01194-f006:**
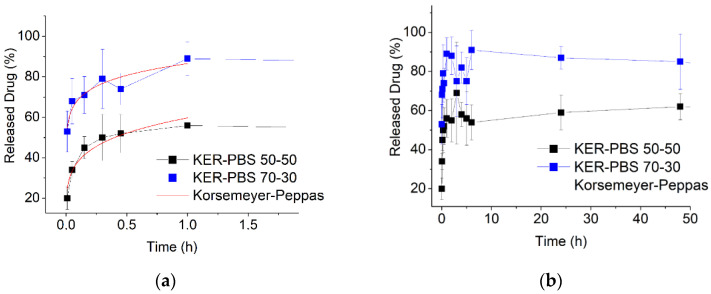
RhB release profiles of Keratin/PBS 50/50 and 70/30 electrospun mats: (**a**) initial RhB release profiles fitted by Korsemeyer-Peppas model, (**b**) complete RhB release profiles.

**Figure 7 biomolecules-11-01194-f007:**
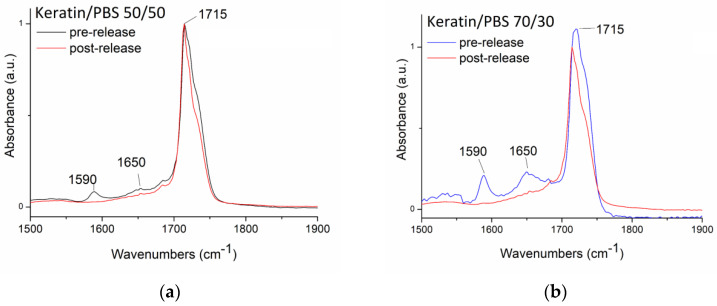
FTIR spectra of (**a**) Keratin/PBS 50/50 and (**b**) Keratin/PBS 70/30 pre and post release.

**Table 1 biomolecules-11-01194-t001:** Carreau model constants of keratin/PBS solutions.

Keratin/PBS	*η*_0_ (Pa s)	$\eta_{\infty }$ (Pa s)	$\lambda_{c}$ (s)	*n*	R^2^
100/0	11 ± 3	0.44 ± 0.05	0.3 ± 0.1	0.42 ± 0.003	0.991
70/30	15 ± 1	0.61 ± 0.01	1.7 ± 0.2	0.44 ± 0.01	0.998
50/50	17 ± 1	0.23 ± 0.02	0.48 ± 0.05	0.444 ± 0.007	0.999
30/70	5.97 ± 0.07	0.55 ± 0.05	2.0 ± 0.2	0.29 ± 0.02	0.986
0/100	1.498 ± 0.08	0.51 ± 0.05	0.022 ± 0.003	0.40 ± 0.08	0.989

**Table 2 biomolecules-11-01194-t002:** Tensile and adhesion tests data of PBS and Keratin/PBS 50/50 electrospun mats.

Keratin/PBS	E (MPa)	σ_B_ (MPa)	ε_B_ (%)	σ_S_ (MPa)	σ_S_ * (MPa)
0/100	20 ± 3	6.2 ± 0.7	151 ± 7	0.7 ± 0.2	n.d.
30/70	16 ± 2	5 ± 1	90 ± 7	n.d.	n.d.
50/50	16 ± 5	3.3 ± 0.3	56 ± 9	1.5 ± 0.3	0.9 ± 0.3

* Samples after drying; n.d.: not-disclosed.

**Table 3 biomolecules-11-01194-t003:** Thermal characterization data (TGA and DSC) of PBS, keratin and Keratin/PBS blend electrospun mats.

Keratin/PBS	T_id_ °C	T_max_ °C	I Scan					
			T_g_ °C	∆C_p_ J/g °C	T_d_ °C	T_m_ °C	∆H_d_ J/g	∆H_m_ J/g
100/0	71 188 271	320	-	-	88 226 282	-	229 48 16	-
70–30	64 214 295	364	-	-	62 220	112	40 7	44
50–50	64 212 309	379	-	-	68 226	111	71 19	27
70–30	205 353	394			72 227	110	90 26	21
PBS	363	398	−32	0.193	-	114	-	64

**Table 4 biomolecules-11-01194-t004:** Kinetic parameters of Korsemeyer-Peppas model.

Keratin/PBS	*k* Min^−*n*^	n	R^2^
50/50	60 ± 3	0.20 ± 0.03	0.924
70/30	86 ± 3	0.10 ± 0.02	0.889

## Data Availability

Not applicable.

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
