# Peer review of "Effects of the Blending Ratio on the Design of Keratin/Poly(butylene succinate) Nanofibers for Drug Delivery Applications"

_biomolecules, 2021, doi:10.3390/biom11081194_

Round 1

Reviewer 1 Report

In the work “Effects of the blending ratio on the design of keratin/poly(buthylene succinate) nanofibers for drug delivery applications” the authors prepared keratin/poly(butylene succinate) blends by using 1,1,1,3,3,3-Hexafluoro-isopropanol as solvent. The blends were electrospunned with rhodamine B as a model to study the ability of the mats as drug delivery systems. The manuscript is very well written, very easy to follow. However, some minor questions should be addressed:

  • Please explain in more detail the novelty of this work when compared with work published previously (Regenerated wool keratin-polybutylene succinate nanofibrous mats for drug delivery and cells culture, doi:10.1016/j.polymdegradstab.2020.109272.).
  • Line 93 – FTIR conditions are missing on methods
  • Line 103, and Line 108 - Why the authors used the concentration of 13% w/V for the preparation of PBS and keratin? Please comment on this.
  • Line 191 – Figure 1 Caption – please include the gel concentration and amount of keratin used.
  • Line 201 – Please check the quality of Eq. 1
  • Line 208 – Please comment on the size and properties of the nanofibers obtained with the Keratin/PBS ratios of 100/0 and 0/100? If these samples were electrospinninable, add the microphotographs and analysis to figure 3.
  • Line 258 – Can the authors comment on the shear stress values (σS) of the Keratin/PBS 30/70 mats?
  • Line 279 – Please add the meaning of # in Table 2
  • Line 298 – Table 3 - Is very hard to read the results of Tid and of ΔHd J/g. Please improve Table 3
  • Line 322 – Please check the quality of Eq. 3

Author Response

The reply to the reviewe 1 is uploaded as word file.

Reviewer 2 Report

Dear editor,

The manuscript “Effects of the blending ratio on the design of keratin/poly(buthylene succinate) nanofibers for drug delivery applications.” describes the preparation and subsequent characterization of nanofibrous membranes from keratin and PBS. Materials are studied to be used as the drug delivery system. The manuscript is written logically and in understandable English, the presented research is very actual. However, there are issues, which should be addressed before possible publication. Therefore, I have the following comments on the submitted manuscript:

INTRODUCTION

  • from the introduction is not entirely clear, why is the advantage of preparing blends in the case of drug-delivery use of nanofibers?
  • furthermore, although the preparation of the blend is described as "widely explored", there are only a few references.
  • there is also a lack of information on pure materials
  • it is not understandable why rhodamine was chosen as a model drug
  • sentence (line 79) “Despite their immiscibility, the blend solutions were successfully electrospun into nanofibrous membranes with good mechanical properties due to the presence of PBS and with stimulatory effect towards fibroblast growth induced by the presence of keratin” should be supplemented by references to the literature

MATERIALS and METHODS

  • for all important used chemicals, the basic description and also the company should be specified incl. place / state
  • electrospinning of…
    • preparation of solutions is not described clearly: keratin / PBS and rhodamine concentrations should be reported as weight percent (w/w). I recommend stating the method of preparation of the solution - how much (mg, volumes) of what was added. It should also be specified whether it is a concentration in solution or in the final material.
    • how / why was the used rhodamine concentration proposed? Was the solution of keratin-rhodamine already solution? Rhodamine solubility is limited ….
    • The electrospinning process should be described in more details incl. info about the used collector, applied current, distance of electrodes, environment conditions…..
  • Characterization
    • this part should be divided and each used method described separately
    • There is no analysis of the composition of final materials– to prove the content and homogeneity of prepared mats (e.g. FTIR, chromatography)
    • Drug release – please specify the time intervals, also calibration curve preparation

RESULTS

  • Keratin regenerated from HIFP
    • analysis of keratin behavior is definitely in place. However, the analysis of keratin is missing for the final material. Not only the solvent, but also the spinning process and the presence of other substances can affect its degradation, so I recommend adding this analysis
    • the analysis of keratin behavior is definitely in place. However, keratin analysis is missing in the final material. Not only the solvent, but also the spinning process and the presence of other substances can affect its degradation, so I recommend adding this analysis
    • the presented electrophoreogram has no proper informative value - the lines are not well assembled (bands of the same molecular weight do not correspond),
    • also, there is strong background - it is not clear that degradation does not occur, so the result cannot be presented in this way
  • material morphology
    • I recommend adding SEM images of all the analysed materials - including pure materials (keratin and PBS) and also materials with incorporated substance
    • complete the analysis of fiber diameters - the resulting average value and the standard deviation, or confidence interval….etc
    • histograms should always be made from the same number of values, so presented ones are not comparable, I also recommend increasing the width of the counts to 100 or at least 50
    • fiber diameters never have a Gaussian distribution - I therefore consider this analysis inappropriate
    • I recommend larger pictures - both SEM and histograms, but also photos in supplementary
    • was the production of the material repeated and the evaluation of the morphology are the results of several sets of materials? If not, then the discussion of morphology is not supported by sufficient data
    • it would also be appropriate to discuss whether the morphology is affected by the incorporated substance
  • drug release
    • I lack rhodamine release analysis from keratin material only
    • what is 100%? Theoretical loading? It is necessary to specify. However, since no analysis has been made of the actual content and the homogeneity of the rhodamine distribution, the theoretical value is only indicative
    • it is not clear whether the rhodamine content was the same for both materials - or was it lower for the material with a lower amount of keratin?
    • I recommend, in addition to the analysis of rhodamine release, also to quantify the released keratin (it will confirm a possible discussion about the formation of the rhodamine-keratin complex)
    • I do not understand the course of release on graph 6b - in the interval of about 100h is there a significant decrease? How can this be explained? If the analysis was performed according to the described procedure (cumulative method), can't it look like this? It can therefore be assumed that the analysis itself or the subsequent processing of the results was not performed correctly.

DISCUSSION

The discussion is based on the results, which must be supplemented and clarified. Also, there is almost no discussion with the results from the literature.

CONCLUSIONS

This part should be improved on the basis of improved results

For the above reasons, I propose the major revision of the article.

Author Response

The reply to Reviewer 2 is uploaded as word  file

Reviewer 3 Report

In this work, the authors developed a novel keratin/PBS blend nanofiber with functional properties tunable by acting on the polymer blending ratios. The keratin/PBS blend nanofibers demonstrated improved the mechanical, thermal and adhesion properties, as well as drug release profile. This work is interesting, and I would like to recommend the acceptance with minor revision.

  1. Does the loading of Rhodamine B affects the viscosity and adhesion ability of the nanofibers.
  2. In Figure 6, the cumulative drug release decreased by 25% from 1 h to 100 h, while it should be increased with time.
  3. 90% or 56% of Rhodamine B was released from the fibers in 1 hour, please explain whether such burst drug release was due to the unstable drug loading or not. And this burst release may conflict with controlled release that this article wish to achieve.

Round 2

Reviewer 2 Report

Dear editor,

All my comments have been taken into account in the revised version of the manuscript “Effects of the blending ratio on the design of keratin/poly(buthylene succinate) nanofibers for drug delivery applications.”  and I have no further comments. I only recommend unifying the scales in fig 3 and increasing the width of the counts at histograms.